# Scale Dependence of Errors in Snow Water Equivalent Simulations Using ERA5 Reanalysis over Alpine Basins

**Susen Shrestha** [1,*] **, Mattia Zaramella** [2]**, Mattia Callegari** [3]**, Felix Greifeneder** [4] **and Marco Borga** [2]

1. Center for Climate Change and Transformation (CCT), Eurac Research, 39100 Bolzano, Italy
2. Department of Land, Environment, Agriculture, and Forestry, University of Padova, 35100 Padova, Italy
3. Institute for Earth Observation, Eurac Research, 39100 Bolzano, Italy
4. Chloris Geospatial, Boston, MA 02116, USA
* Correspondence: sshrestha@eurac.edu

**Abstract:** This study aims to evaluate the potential of ERA5 precipitation and temperature reanalysis for snow water equivalent (SWE) simulation by considering the role of catchment spatial scale in controlling the errors obtained by comparison with corresponding SWE simulations from ground stations. This is obtained by exploiting a semi-distributed snowpack model (TOPMELT) implemented over the upper Adige River basin in the Eastern Italian Alps, where 16 sub-catchments of varying sizes are considered. The comparison is carried out from 1992 to 2019. The findings show that ERA5 precipitation overestimated low-intensity rainfall (drizzle problem) and underestimated high-intensity rainfall, while ERA5 temperature underestimated observations. The overestimation of low-intensity rainfall created fictitious low-intensity snowfall events, which, when combined with colder ERA5 temperature, resulted in delayed snowmelt and increased fictitious snow-cover days over the study area. The quantile mapping (QM) technique was used to remove errors in ERA5 variables. It was shown that ERA5 could struggle to resolve the orographic enhancement in precipitation, which may be particularly important during high-SWE years. This reduces the positive precipitation bias during those years, thus reducing comparatively the ability of the quantile mapping technique to correct for bias homogeneously during all years. This study highlighted the importance of temperature correction over precipitation correction in SWE simulation, particularly for smaller basins.

**Keywords:** TOPMELT; ERA5; seasonal snowpack simulation; snowmelt; drizzle problem; bias adjustment

## 1. Introduction

Seasonal mountain snowpack and its melt dominate the surface hydrology of many regions, with implications for water supply, hydropower, risk assessment, and ecological processes [1–4]. Reliable snow-cover (SC) delineation and snow water equivalent (SWE) assessment remains crucial for snowmelt runoff prediction, operational flood control, water supply planning, and water resource management in snowmelt-dominated watersheds [5]. However, the proper assessment of SC and SWE in mountain watersheds still remains a challenge. In data-scarce basins, the use of reanalysis data, such as ERA5 [6], is of great interest to the hydrological community. Reanalysis data combine a wide array of measured and remotely sensed information within a dynamical–physical coupled numerical model. They use the analysis part of a weather forecasting model, in which data assimilation forces the model toward the closest possible current state of the atmosphere [7]. Of particular interest to the hydrological community are the largely improved (with respect to earlier reanalysis products) spatial (30 km) and temporal (1 h) resolutions of ERA5. The spatial resolution is now similar to or better than that of most observational networks in the world, with the exception of some parts of Europe and the United States. In particular, the ERA5 precipitation dataset was found to reproduce the pattern of the daily precipitation climate

well in three European regions (Alps, Carpathians, Fennoscandia), with reproduction of contrasts in mean precipitation between mountains and flatland and gradients of wet-day frequency across crests [8].

In spite of the interest, the use of ERA5 data to drive snowpack models has not received large attention in the relevant literature. In two studies [9,10], ERA5 has been shown to provide reasonable simulations of SWE for mountain basins. The performance of MERRA-2 and ERA-5 reanalysis meteorological forcing was compared with the distributed SnowModel in the high Atlas region [9]. Both the reanalysis products showed comparable performance for snow simulation. Nonetheless, the ERA5 simulation, due to its finer spatial resolution as compared to MERRA-2, showed better performance, whereas [10] considered ERA5 as the boundary and initial condition to force the WRF model for two mountain ranges in Lebanon. The Intermediate Complexity Atmospheric Research Model (ICAR) was nested into the WRF to develop a 1 km regional-scale snow reanalysis. The results showed a good temporal and spatial correlation of the snow variables with the MODIS fractional snow-covered area and the ground observations of SWE.

In these studies, the ERA5 spatial resolution has been shown to be still too coarse to correctly represent the influence of topography on meteorological variables, which is crucial for snow modeling in mountainous regions [10–12].

Whereas a number of studies proposed downscaling techniques to approach this problem, the scale-dependent structure of errors in SWE simulations using ERA5 reanalysis over alpine basins remains largely unexplored. This is in spite of the need to better understand at which spatial scales the ERA5 spatial resolution mostly affects the SWE simulations.

The scale effect is an important topic and belongs to 1 of the 23 unsolved problems in hydrology [13]. Namely, what are the hydrological laws at the catchment scale, and how do they change with scale? In this study, we attempt to answer the following questions: What is the spatial representation of ERA5 precipitation and temperature as forcing data for SWE simulation at the catchment scale? Furthermore, how do they change with scale?

The aim of this work is to evaluate the quality of hourly temperature and precipitation data from ERA5 reanalysis to simulate SWE dynamics in a mountainous, snow-controlled river system, with respect to corresponding SWE simulations obtained from a relatively high-density and quality-controlled data set obtained from ground stations, used as a reference.

Therefore, this study aims (i) to isolate the impact of the input spatial aggregation on the accuracy of SWE simulations by quantifying the effects of aggregating the reference precipitation and temperature data at the ERA5 grid-scale and (ii) to evaluate the scale-dependence of ERA5-based simulation errors when SWE is aggregated over a range of spatial scales. To allow these analyses, mean areal precipitation estimates and temperature will be generated at the ERA5 spatial resolution using station precipitation data. This input data is referred to as Station Input at ERA5 Resolution (SIER). Then, two sets of comparisons will be carried out: (i) SIER vs. a reference based on station data to isolate the genuine impact of the aggregation of input data at the ERA5 resolution; and (ii) ERA5 vs. SIER to evaluate the impact of ERA5-only on SWE simulation, therefore disregarding the pure effect of spatial resolution.

This study was carried out in the upper Adige River basin, a 6924 km$^2$ wide basin located in the Eastern Italian Alps, over the 1992–2019 period. The study area was selected due to the dense network of meteorological stations used to provide input to a snowpack model and to validate the ERA-5 performance. The TOPMELT model [14] is used for seasonal snowpack dynamics and SWE simulation. TOPMELT is a semi-distributed snowpack model based on an extended temperature index approach capable of estimating the full spatial distribution of the SWE at each time step. TOPMELT exploits a statistical representation of the distribution of clear-sky potential solar radiation to drive the snowpack model, which drastically reduces the computational costs associated with the fully spatially distributed simulation of SWE over vast areas and an extended period of time while preserving simulation accuracy [14]. The good accuracy of TOPMELT SWE and SC

simulations over the study area has been tested with respect to available in situ data and MODIS observations by [15,16].

## 2. Materials and Methods

### 2.1. Study Area and In Situ Data

The study area is represented by the upper Adige River basin closed at Bronzolo in the Eastern Italian Alps (Figure 1). This is an alpine catchment with a drainage area of approximately 6924 km$^2$. The elevation ranges from about 200 m a.s.l. in the southern valley bottoms to around 3900 m a.s.l. in the western upper ranges, with a mean elevation of 1800 m a.s.l.

The steep terrain and the high elevation gradients govern the spatial precipitation distribution [17], with the precipitation ranging from 500 mm in the northwest region to 1600 mm in the north-central region [18]. During the winter season, the precipitation is stored as snow, and the streamflow is minimum. The streamflow shows two maxima: the first due to snowmelt in the early summer and the second due to intense precipitation in autumn [19]. The main agricultural areas in the northern part of the catchment are located in the Venosta valleys, and cultivation comprises mainly fruit trees and grapes. Land use at high elevations is dominated by grass, grazing, and forest.

The study area is characterized by a rather dense network of meteorological stations, with 88 rain gauges (1 per 72 km$^2$) and 124 temperature gauges (1 per 55 km$^2$) covering the study region. The Hydrographic Office of Bozen, Bolzano, has made hourly temperature, precipitation, and runoff data available from 1991 until 2019. To assess the performance of the model at varying basin sizes, sixteen sub-catchments within the study basin were selected for the analysis. The drainage areas of the study basins range from 49 km$^2$ to 6924 km$^2$ (Figure 1, Table 1). These watersheds were chosen for analysis due to the relatively minor impact of operations from artificial reservoirs, which allowed for the use of a hydrological model and ensuing comparison of simulated versus observed discharges.

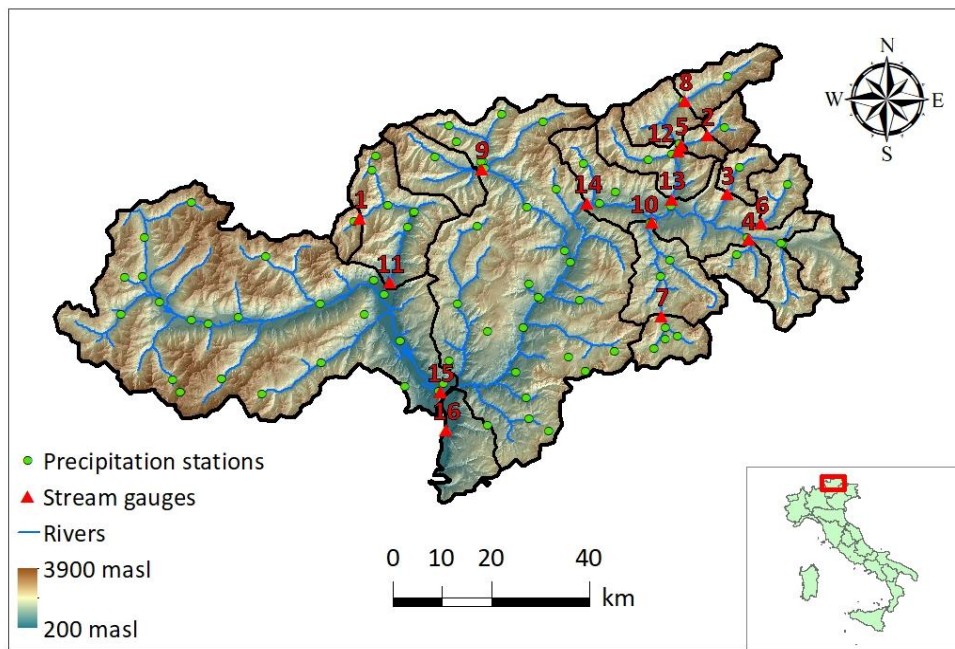

**Figure 1.** The upper Adige River basin closed at Bronzolo. The red triangles indicate the stream gauges, which are listed in Table 1.

**Table 1.** Drainage area and elevation of the study basins.

| Sn | Name | Elevation Range (m) | Mean Elevation (m) | Area (km²) |
|---|---|---|---|---|
| 1 | Rio Plan at Plan | 1561–3445 | 2387 | 49 |
| 2 | Rio Riva at Seghe | 1523–3421 | 2386 | 76 |
| 3 | Rio Anterselva at Bagni | 1092–3421 | 2026 | 82 |
| 4 | Rio Braies at Braies | 1124–3074 | 1911 | 93 |
| 5 | Rio Riva at Caminata | 855–3421 | 2278 | 115 |
| 6 | Rio Casies at Colle | 1196–2815 | 1961 | 117 |
| 7 | Rio Gadera at Pedraces | 1318–3111 | 2027 | 125 |
| 8 | Aurino at Cadipietra | 811–3111 | 2162 | 150 |
| 9 | Rio Ridanna at Vipiteno | 1046–3417 | 1933 | 210 |
| 10 | Gadera at Mantana | 944–3441 | 1855 | 397 |
| 11 | Rio Passirio at Merano | 336–3445 | 1851 | 414 |
| 12 | Aurino at Caminata | 844–3421 | 2117 | 420 |
| 13 | Aurino at S.Giorgio | 817–3421 | 2036 | 608 |
| 14 | Rienza at Vandoies | 732–3421 | 1859 | 1919 |
| 15 | Adige at Ponte Adige | 236–3889 | 1895 | 2732 |
| 16 | Adige at Bronzolo | 236–3889 | 1805 | 6924 |

*2.2. ERA5 Reanalysis*

ERA5 reanalysis is a state-of-the-art fifth-generation ECMWF (European Centre for Medium-Range Weather Forecasts) atmospheric reanalysis of the global climate [6]. It is one of the fundamental elements of the Copernicus Climate Change Service (C3S), which is funded by the European Union. ERA5 provides multiple atmospheric, land, and oceanic climate data variables with availability spanning from 1959 to the present at a spatial resolution of 0.25 degrees and a temporal resolution of 1 h at the global scale. For this research, only temperature and precipitation will be considered from ERA5. Further information about ERA5 is available on the online data documentation (https://confluence.ecmwf.int/display/CKB, accessed on 5 May 2023). It provides a detailed description of the various products and a list of all available geophysical parameters, which can be freely downloaded.

Twenty-seven ERA5 grid cells that cover the upper Adige River basin at Bronzolo were considered for this study. Based on geometrical analysis, the ERA5 precipitation was partitioned over the 16 study basins. The ERA5 air temperature is scaled to the center of mass of each study basin based on the climatological monthly lapse rate valid for the region.

*2.3. The Snowpack Model: TOPMELT*

The snowpack model used in this work is TOPMELT [14]. TOPMELT is a semi-distributed snowpack model which takes advantage of the extended temperature index approach to simulate SWE at full spatial distribution for each hourly time step [10,12]. Clear-sky shortwave solar radiation is computed at each element of the digital terrain model (DTM) by taking into account shadow and complex topography, calculating the apparent sun motion, and the intersection of radiation with topography. Diffuse radiation is computed by accounting for self-shading (by slope and aspect) and occlusions produced by the visible horizon. For model simulation, each basin is divided into $n_b$ elevation bands, and the full spatial distribution of clear-sky potential solar radiation is discretized into a number of radiation classes for each elevation band. This is achieved by dividing each elevation band into $n_c$ equally distributed radiation classes, where the $i^{th}$ class contains the band

sub-area corresponding to the $i^{th}$ percentile of the incident radiation energy. TOPMELT accounts separately for snow and glacier melt; to account for the presence of a glacier area associated with an energy class, each one of the $n_b \times n_c$ model cells is characterized by the corresponding fraction of glacier area.

Snowpack simulation is carried out for each radiation class rather than for each DTM pixel, which significantly improves computational efficiency. The analysis by [14] showed that the division of the elevation bands into ten radiation classes provides results comparable to a full spatially distributed model. Therefore, in this work, the elevation bands are subdivided into ten radiation classes based on the spatial distribution of the clear-sky solar radiation of the pixels included in each elevation band.

The model uses the Thiessen polygon method to estimate the mean precipitation over the basin, while the air temperature data is used to calculate the hourly temperature lapse rate for the basin. The precipitation data are corrected with the snow correction factor (*SCF*) to account for gauge catch deficiencies during snowfall. The *SCF* is a multiplier applied to precipitation data when the station temperature goes below the threshold temperature $T_c$, which identifies solid precipitation. Lastly, the basin precipitation, $p_{basin}$, is obtained by applying a precipitation correction factor (*PCF*), which is a non-dimensional constant used to take into account the poor spatial coverage of the rain gauge stations. For a given $i^{th}$ elevation band, the model computes the precipitation $p_i(t)$ (mm h$^{-1}$) at time $t$ by applying a vertical precipitation gradient $G$ (km$^{-1}$), which considers increased precipitation over elevation as given in Equation (1):

$$P_i(t) = P_{basin}(t)\left(1 + G\frac{h_i - h_{ref}}{1000}\right) \tag{1}$$

where $h_i$, $h_{ref}$ (m a.s.l) are the mean altitude of the $i^{th}$ elevation band and of the basin, respectively. For the temperature, $T_i$ (°C) is provided as input for each time step and elevation band. The use of a vertical temperature lapse rate helps to obtain the mean air temperature over the $i^{th}$ elevation. The estimation of the precipitation phase (solid or liquid) is performed using the threshold temperature $T_c$, thus obtaining the snow water equivalent of the precipitation at time $t$, $snow_i(t)$.

The snowmelt algorithm is applied to each radiation class of a given elevation band. For a given $i^{th}$ elevation band and $j^{th}$ radiation class, the snowmelt rate $F_{i,j}(t)\left[\text{mmh}^{-1}\right]$ at time $t$ is calculated as in Equation (2):

$$F_{i,j}(t) = CMF \cdot RI_{i,j}(t) \cdot (1 - alb_i(t)) \cdot max\{0; T_i(t) - T_b\} \tag{2}$$

Here, *CMF* is the combined melting factor considering both thermal and radiative effects, $RI_{i,j}(t)$ is the cell radiation index, $alb_i(t)$ is the snow albedo [-] at time $t$, $T_i(t)$ is the air temperature at the $i^{th}$ elevation band at time $t$, while $T_b = 0$ °C is the temperature threshold above which snowmelt is assumed to occur, both in [°C]. The snow albedo at each elevation band is computed using the method described by [20] as given in Equation (3):

$$alb(t) = ALBS - \beta_2 \cdot [log_{10}\sum_k T_i(t_k)] \tag{3}$$

where *ALBS* is the albedo of fresh snow, $\beta_2$ is a dimensionless parameter, and $\sum_k T_i(t_k)(°C)$ is the summation of the positive hourly temperatures that are above the threshold base temperature ($T_b$) from the last snowfall until the current time $t$. The model accounts for rain-on-snow and melting during the night employing a temperature index approach through two additional parameters: the rain melt factor (*RMF*) and the night melt factor (*NMF*).

The SWE ($we_{i,j}$, mm) for each model cell is updated considering the snow water equivalent of the precipitation and melt, as given in Equation (4):

$$we_{i,j}(t) = we_{i,j}(t-1) + snow_i(t) - F_{i,j}(t) \tag{4}$$

The model also computes ice melt. When we$_{i,j}$ is less than a threshold (*WETH*), ice melt begins. The glacier melt is computed as given in Equation (2), but the snow albedo is replaced with a constant glacier albedo (*ALBG*).

TOPMELT can simulate the full spatial distribution of the SWE; hence, the output from the TOPMELT can be compared against the point ground observations and also with snow-cover products like MODIS. The snow-cover area (SCA) is calculated here by considering a pixel as snow-covered when the simulated SWE exceeds a threshold of 5 mm, based on [16].

## 2.4. The Hydrological Model: ICHYMOD

The ICHYMOD hydrological model, developed by [21], combines the TOPMELT snowpack model from [14] with a conceptual rainfall–runoff hydrological model at the basin scale. This model converts snowmelt and excess precipitation into runoff at the basin outlet and includes a snow routine, soil moisture routine, and flow routine. The soil moisture routine uses the probability-distributed model (PDM) developed by [22] to describe the spatial distribution of water storage capacity across the basin. The cubic law storage model is used to route base discharge from groundwater to the catchment outlet. The Hargreaves method [23] is used to compute losses due to evapotranspiration, taking into account the status of soil moisture stored in the PDM. The model represents fast and slow response pathways using storage-based representations and calculates the total basin flow by summing the spatially lumped representations of fast and slow response at the outlet. Drainage to the slow flow path is a function of basin moisture storage, and the slow or base flow component of the total runoff is routed through an exponential store. Direct runoff from areas where storage capacity is exceeded is routed through a geomorphology-based distributed unit hydrograph, using a geomorphologic filter based on a threshold drainage area to distinguish hillslopes and channel networks. The routing time for each site in the basin is evaluated by assigning different typical velocity values to each pixel and classifying them as either hillslope or channel flow, with two velocities used to describe the flow routing.

As the TOPMELT model operates in conjunction with the ICHYMOD model, the model parameters for TOPMELT were calibrated based on runoff simulations at 16 river gauge stations, each corresponding to a sub-catchment outlet as given in Table 1 and MODIS snow-cover data as in [16].

## 2.5. ERA5 Bias Adjustment Method

To simulate the regional snow dynamics realistically, it is necessary to correct the ERA5 precipitation and temperature for biases. For this, a quantile-quantile mapping (QM) transformation [24] is used here for precipitation and temperature. The implemented QM scheme was based on the R package qmap [25]. For a given variable, the cumulative density function (CDF) of ERA5 is first matched with the CDF of the references, generating a correction function depending on the quantile. Then, this correction function is used to unbias the ERA5 variable quantile by quantile. QM was applied separately for each month. To avoid overfitting due to the small sample size of monthly values included in the calibration, the quantile adjustment was computed by considering deciles instead of centiles and applied by linearly interpolating the empirical distribution. For precipitation, a wet-day correction equalizing the fraction of days with precipitation between the observed and the modeled data was applied. The correction was applied to each cell separately. For precipitation, the procedure was calibrated over a calibration period (1992–2005). A validation period (2005–2019) was used to examine the quality of the correction scheme.

Examination of the temperature data showed a step change in 2005, confirmed by using the Pettitt test [26]. Based on this evidence, the QM procedure was applied separately for the two periods, namely, before and after 2005.

In the following, three QM-corrected ERA5 inputs are considered: one where only precipitation is corrected (termed precipitation-corrected ERA5, PC-ERA5), another one

with only the temperature correction (termed temperature-corrected ERA5, TC-ERA5), and lastly, where both precipitation and temperature are corrected (termed precipitation–temperature-corrected ERA5, PTC-ERA5). Additionally, for the case of temperature, a blind test to evaluate the effectiveness of using bias correction from the calibration period over the whole period is reported. This is termed PTC-cali-ERA5. The results of these comparisons are limited to SWE error calculations for the validation period.

### 2.6. Reference Precipitation at ERA5 Spatial Resolution

To investigate the scale-dependence character of ERA5 errors, mean areal precipitation estimates were generated at the ERA5 spatial resolution using station precipitation data. This input data was referred to as Station Input at ERA5 Resolution (SIER). The Thiessen polygon method was used to redistribute the observed hourly station precipitation data over each ERA5 grid footprint. For temperature, the observed dataset was used without any modifications. The availability of SIER estimates permitted to obtain two sets of comparisons: (i) SIER vs. the reference based on station data to isolate the genuine impact of the aggregation of input data at the ERA5 resolution; and (ii) ERA5 vs. SIER to evaluate the impact of ERA5-only on SWE simulation, therefore disregarding the pure effect of spatial resolution.

### 2.7. Comparison Statistics

The comparison is carried out using the Kling–Gupta efficiency (KGE) [27] as calculated using Equation (5):

$$\text{KGE} = 1 - \sqrt{(\gamma - 1)^2 + (\beta - 1)^2 + (\alpha - 1)^2} \tag{5}$$

and the mean bias ratio (MBR) as given in Equation (6).

$$\beta = \frac{\overline{Estimated}}{\overline{Reference}} \tag{6}$$

Here, $\gamma$ represents the correlation component, represented by Pearson's correlation coefficient; $\beta$ is the mean bias ratio, represented by the ratio of estimated and reference means, respectively indicated as $\overline{Estimated}$ and $\overline{Reference}$ in Equation (6); and $\alpha$ is the variability component, represented by the ratio of the estimated and reference coefficients of variation. KGE ranges from negative infinity to one, where a value of one indicates a perfect match between the two series. The Kling–Gupta efficiency (KGE) is calculated based on all data points except intervals where both data sources are zero.

## 3. Results

Figure 2a,b report the time series of annual mean values of temperature and precipitation, respectively, for the Adige River basin closed at Bronzolo, reporting both the reference and ERA5 values. An overall bias of 1.36 affects the annual ERA5 precipitation totals, resulting from over-prediction of smaller precipitation and under-prediction of larger precipitation events. In contrast, ERA5 temperature data exhibits a non-stationary behavior, with two different cold biases observed for the first and second half of the data. Applying the Pettitt test to detect changes in ERA5 temperature, it showed a change point in 2005, with a bias of $-0.91$ °C for the 1991–2005 period and a bias of $-0.29$ °C for the 2005–2019 period.

Figure 3a,b shows a comparison of uncorrected and QM-corrected ERA5 precipitation for the calibration period. Hourly mean areal precipitation data of the 16 study basins are considered. Figure 3c,d show the same comparison for the validation period. The KGE values of ERA5 precipitation for both periods indicate similar performances, which highlights the robustness of the QM procedure employed in this study.

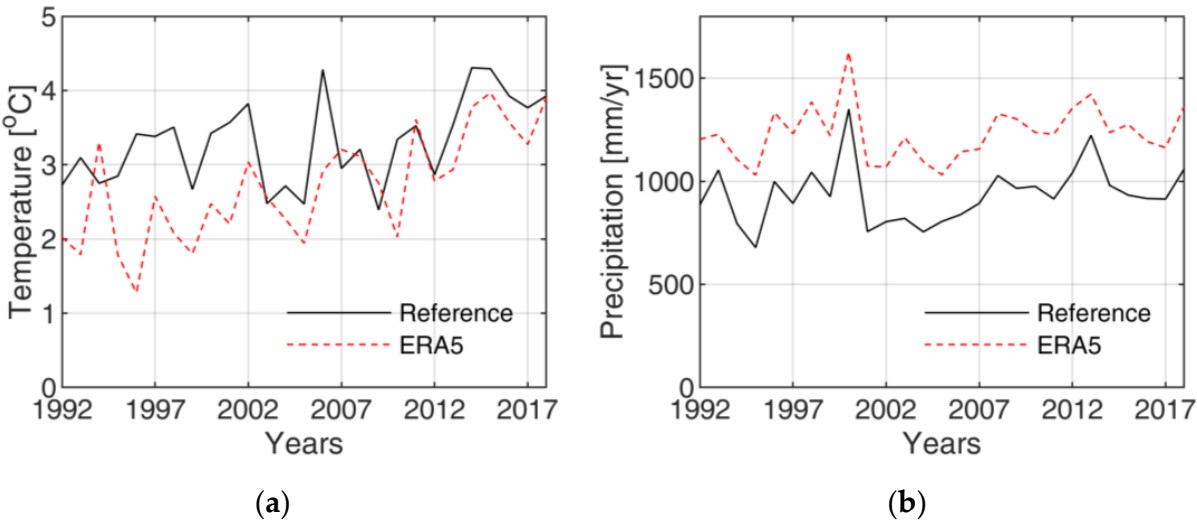

**Figure 2.** Adige at Bronzolo: mean annual (**a**) temperature and (**b**) precipitation.

(**a**)

(**b**)

(**c**)

(**d**)

**Figure 3.** KGE of precipitation: (**a**) ERA5-calibration, (**b**) PC-ERA5-calibration, (**c**) ERA5-validation, (**d**) PC-ERA5-validation.

The KGE shows a clear increasing trend with the basin size in all cases, with the smallest basin (Rio Plan at Plan) exhibiting a KGE of less than 0.2 and the largest basin (Adige at Bronzolo) showing a KGE of just over 0.5. The QM technique applied to ERA5 precipitation reduces the bias of the ERA5 precipitation while maintaining the correlation intact. However, the overestimation of the variability of the observations only results in a marginal improvement in the KGE of PC-ERA5.

Figure 4a,b report the KGE for SIER precipitation for the calibration and validation period, respectively, as compared with the observation. As expected, the figure shows a clear scale dependence in the errors generated by aggregating the reference precipitation at the ERA5 resolution. For basins larger than 1000 km$^2$, the KGE is close to one, whereas it decreases markedly for basins smaller than 1000 km$^2$ and even more remarkably for basins less than 100 km$^2$, with values around 0.6 for the smallest basin.

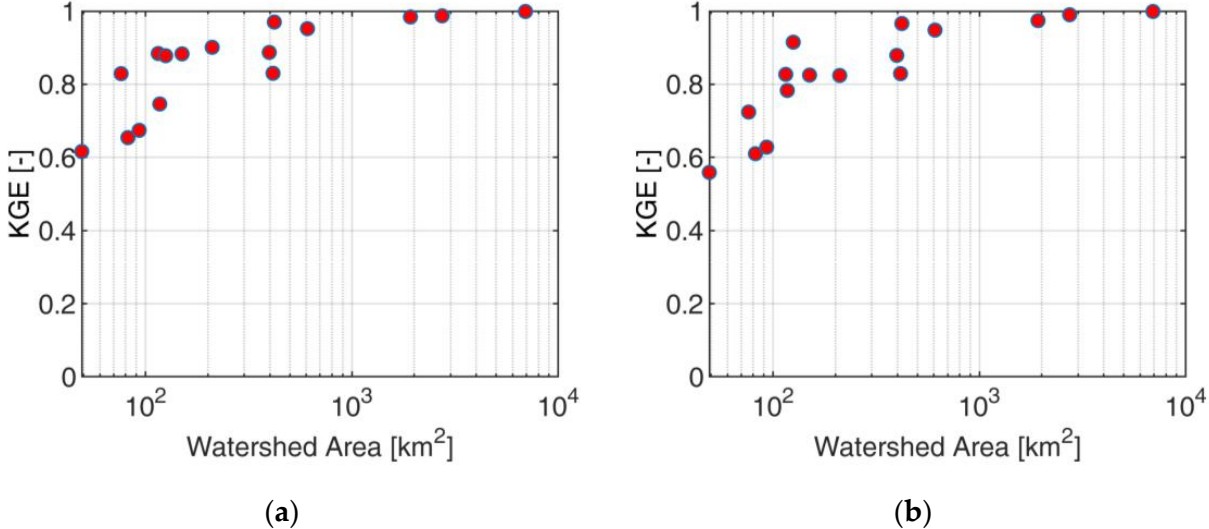

(**a**)                    (**b**)

**Figure 4.** KGE of SIER precipitation with station: (**a**) calibration and (**b**) validation.

Figure 5a,b show the SWE simulations for the whole period for Braies and Adige at Bronzolo, respectively, using three different inputs: reference, ERA5, and PTC-ERA5. Braies is selected because it shows the worst performance (KGE = −0.351) for the ERA5 SWE simulation over the entire study period. The Adige at Bronzolo is selected as it represents the whole study basin.

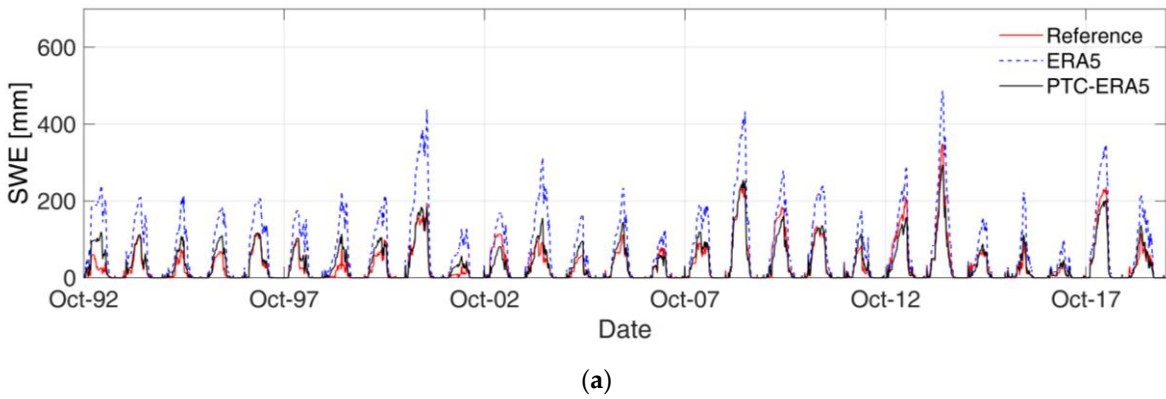

(**a**)

**Figure 5.** *Cont.*

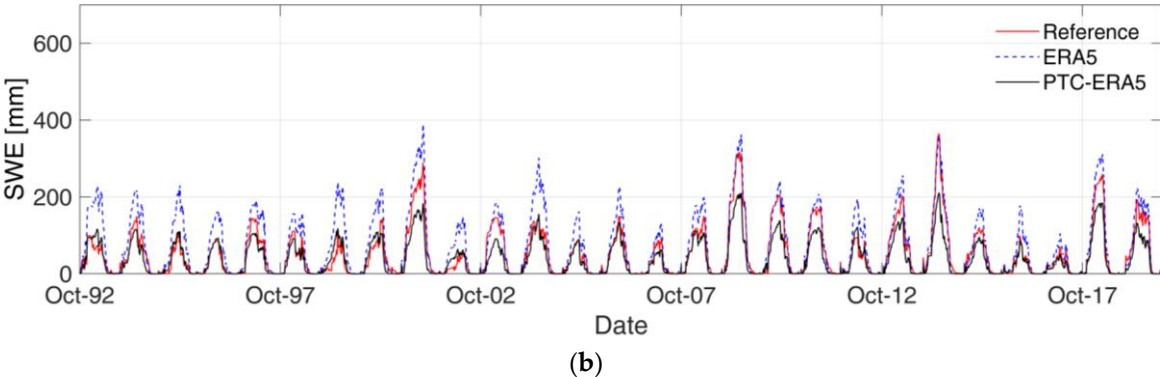

(**b**)

**Figure 5.** SWE time series for different inputs for (**a**) Braies and (**b**) Adige at Bronzolo.

It is evident for both basins that ERA5 overpredicts the SWE, as expected, due to its positive precipitation bias and the cold temperature bias. The ERA5 drizzle precipitation produces smaller false snowfall events, and when coupled with the lower temperature from ERA5, the resulting SWE simulation is overestimated for longer periods. The QM correction for ERA5 precipitation helps to reduce the wet bias. For PTC-ERA5, the results are relatively good for both basins (see Table 2, which provides a summary of statistical comparison). However, a large SWE underestimation is reported after ERA5 correction for Adige at Bronzolo, which also reduces KGE with respect to Braies. The figure clearly shows that the main problem with the SWE underestimation after precipitation and temperature correction at Bronzolo is linked to the nature of ERA5 error, with an overestimation that decreases with SWE accumulation. The high SWE years are generally well represented by ERA5 at Bronzolo, whereas low SWE years are generally overestimated. In Braies, the ERA5 error is much more stable over all the years.

**Table 2.** Statistics considering ERA5 and PTC-ERA5 SWE.

|  | ERA5 | | PTC-ERA5 | |
|---|---|---|---|---|
| **Statistics** | **Braies** | **Bronzolo** | **Braies** | **Bronzolo** |
| Precipitation MBR | 1.31 | 1.36 | 0.96 | 0.95 |
| Mean temperature difference (°C) | −0.56 | −0.57 | 0.02 | 0.02 |
| SWE MBR | 2.10 | 1.44 | 1.06 | 0.75 |
| SWE KGE | −0.35 | 0.47 | 0.91 | 0.62 |

This explains the different efficiency of the ERA5 correction in the two basins. For Braies, the correction is very efficient over all the years, whereas for Bronzolo (as well as for most of the study catchments), the correction is performing well only for low SWE years and leads to relatively strong underestimation for high SWE years.

Figure 6 shows the mean bias ratio and KGE performance of SWE for the different inputs during the validation period. For ERA5 input, the mean bias ratio ranges from 0.9 to 1.8 for the small basins and converges to 1.3 for the larger basins, with an overall overestimation, as discussed previously for Figure 5a,b. The structure of this error translates in the structure of KGE, which shows KGE ranging from 0.1 and 0.95 for small basins and converging to 0.72 for Adige at Bronzolo.

The application of the precipitation-only correction leads to a significant underestimation, with the MBR ranging from 0.55 to 1.05 for the small basins and converging to 0.76 for Bronzolo. Even in this case, the bias controls the overall performance described by the KGE, which is, in general, lower than for the original ERA5 input. Indeed, the KGE ranges from 0.4 and 0.95 for small basins and converges to 0.6 for Adige at Bronzolo, where it is 0.72 for ERA5. These results are due to the difficulties of the QM technique to correct equally well for positive bias for high SWE and low SWE years. As shown before in Figure 5, ERA5 shows increased bias for years with decreasing SWE, and the QM correction is typically

unable to perform well for years with high SWE. This leads to high underestimation in those years, which translates into relatively low KGE performances. In spite of this, the correction is very good in improving the very bad performances for some small basins, where ERA5 performances were also bad in high SWE years (such as for Braies).

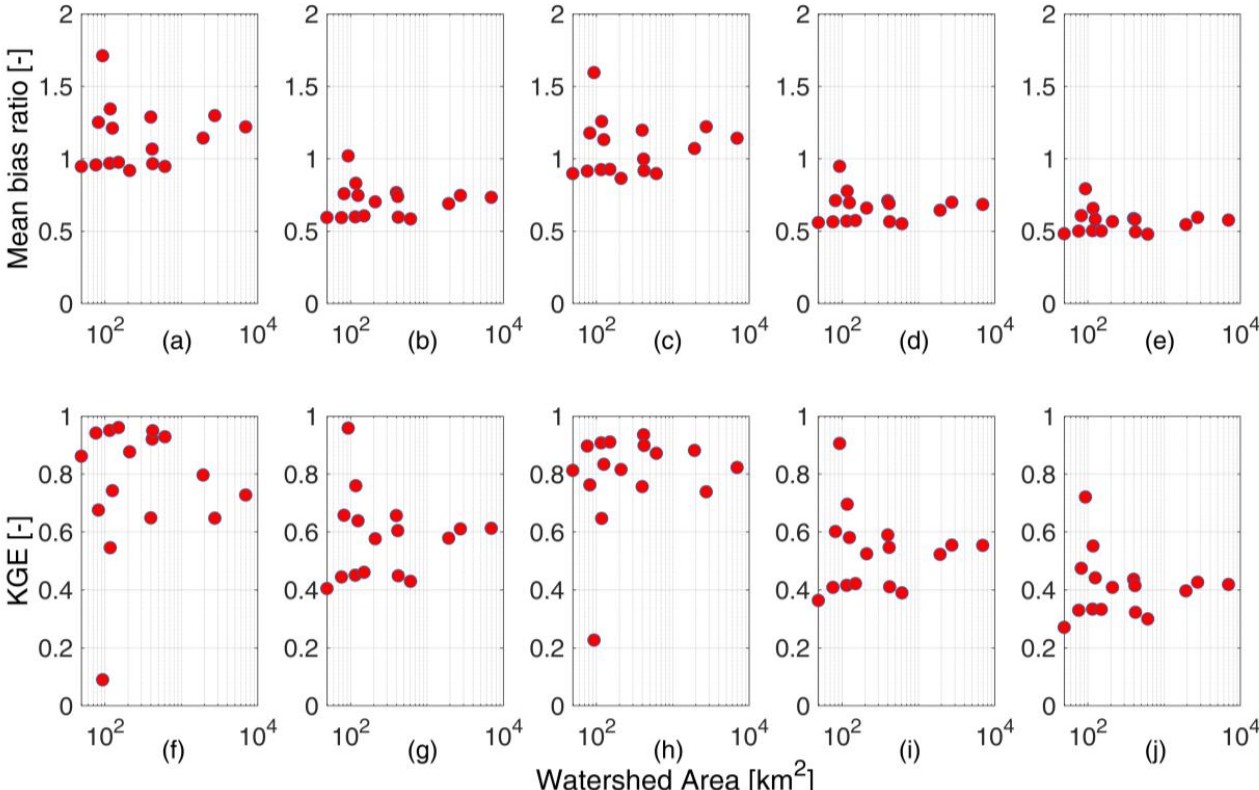

**Figure 6.** SWE simulation performances over the validation period: mean bias ratio for (**a**) ERA5, (**b**) PC-ERA5, (**c**) TC-ERA5, (**d**) PTC-ERA5, (**e**) PTC-cali-ERA5; and KGE for (**f**) ERA5, (**g**) PC-ERA5, (**h**) TC-ERA5, (**i**) PTC-ERA5, (**j**) PTC-cali-ERA5.

While the temperature is solely corrected, there is a significant improvement in both the mean bias ratio and KGE, which highlights the importance of temperature for snow simulation at all scales.

However, the SWE simulation performance, when both temperature and precipitation are corrected, is slightly decreasing as compared to the performance obtained with the precipitation-only correction (and strongly decreasing with respect to the temperature-only correction). This is obviously related to the correction for cold bias ERA5, which adds to the overall underestimation obtained with precipitation-only correction.

This is finally echoed in the results obtained when both temperature and precipitation are corrected, but temperature correction from the calibration period is transferred to the validation period. In this case, a more important correction for temperature is applied, which further amplifies the overall underestimation.

Figure 7 displays the mean bias ratio and KGE of SWE simulation with SIER input for the validation period compared with the reference SWE considering observation. As seen in Figure 4 with the performance of SIER precipitation, the SIER SWE simulation is strongly controlled by the basin scale, with performances for small basins down to 0.41 for the worst case. This shows that for basins smaller than 100 km$^2$, the impact of basin-scale errors may be in the same range as ERA5-only errors.

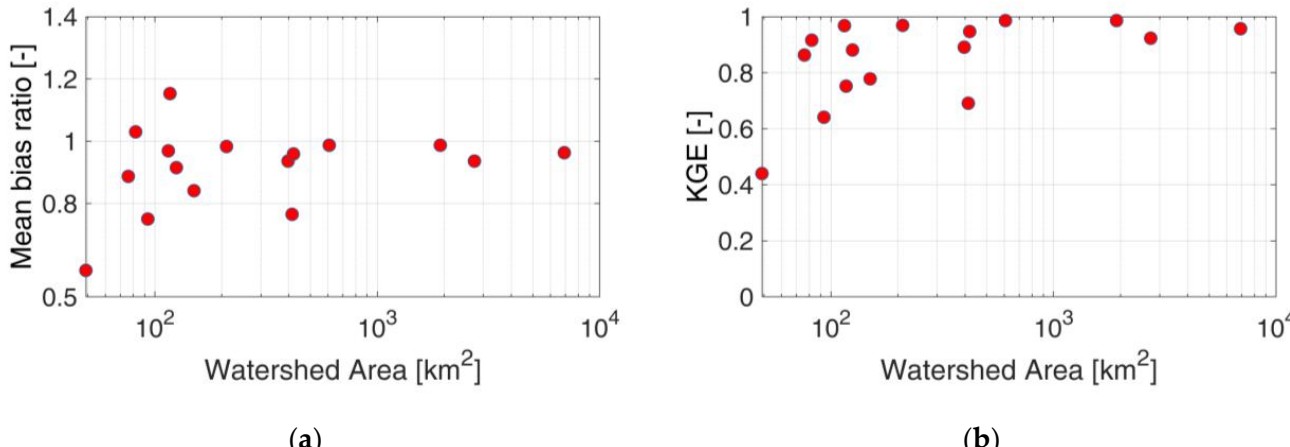

(**a**)                                                                                      (**b**)

**Figure 7.** SWE simulation performances obtained by using SIER input over the validation period: (**a**) mean bias ratio and (**b**) KGE.

Figure 8 reports the fractional snow-cover area (FSCA) simulation for the Adige at Bronzolo for the different inputs considered. FSCA is the percentage of the overall study area covered by snow and helps to understand the impact of various inputs on the spatial snow distribution. The FSCA for ERA5 input simulation is exaggerated, primarily due to the drizzle problem of ERA5 precipitation. As there is a significant number of false precipitation events in ERA5, it artificially increases snowfall in the basin. The lower temperature of ERA5 also delays the melting process, falsely increasing the number of snow-cover days. For PTC-ERA5, the FSCA resembles the reference, and the number of days with 100% FSCA is comparable to the reference.

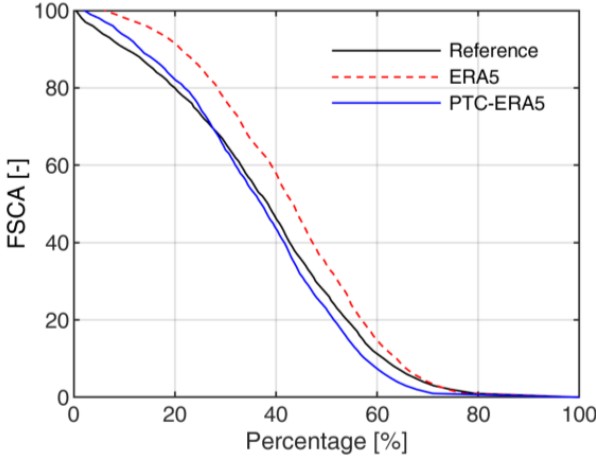

**Figure 8.** FSCA of Adige at Bronzolo for different inputs.

## 4. Discussion and Conclusions

This study presents a comprehensive evaluation of ERA5 meteorological forcing for simulating SWE using the TOPMELT snowpack model in an alpine catchment from 1992 to 2019. The initial findings reveal a positive bias in ERA5 precipitation resulting from lower precipitation events and a cold ERA5 temperature bias compared to observations. This study employs SIER as a reference to comprehend the impact of ERA5 spatial scale on both precipitation and SWE simulation. SIER results show that ERA5's spatial scale affects only smaller basins, whereas the effect is negligible for larger basins. Both the precipitation and SWE performance of SIER decline for basin sizes below 1000 km$^2$, with a distinct effect observed for basins with an area less than 100 km$^2$.

The precipitation bias in ERA5 causes the overestimation of snow, while lower temperature further delays the melt in the study area. The ERA5 temperature bias's impact is

evident in SWE calibration and validation periods. As shown by [28], ERA5 can struggle to resolve the orographic enhancement in precipitation, which may be particularly important during high-SWE years. This reduces the positive precipitation bias during those years, thus reducing comparatively the ability of the quantile mapping technique to correct for bias homogeneously during all years. In some basins, too much ERA5 precipitation can occur on the leeward side of an orographic barrier, as was shown for Braies. This leads to a more homogeneous positive bias across the years and results in very good corrections for bias by means of the quantile mapping technique. However, these positive outcomes are restricted to a few basins. A majority of the study basins show a reduction in SWE estimation performances with the application of the quantile mapping technique.

The best results are thus obtained when only temperature bias correction is applied. Temperature correction significantly improves the KGE performance for a range of basin sizes, with smaller basins showing comparatively greater performance enhancement. Although the temperature's role in SWE is evident regardless of basin size, it should be noted that temperature is more evident for SWE simulation in smaller basins. This is due to their more limited extension in altitude and, therefore, to the temperature effect in identifying the precipitation either as solid or liquid. The blind test of temperature correction highlights the importance of applying different correction factors for different periods, as ERA5 displays distinct temperature behavior in the study period's first and second halves. As small temperature variations can significantly alter the snow process in high-altitude, smaller catchments, proper correction is recommended not only for precipitation but, most importantly, for ERA5's temperature dataset.

The methodology developed in this study can be easily adapted to other similar watersheds with slight modifications to the TOPMELT model. This work also creates opportunities to explore the use of the ERA5 dataset with other modeling approaches for simulating various hydrological processes. As a continuation of this study, the researchers plan to investigate the performance of the ERA5 dataset along with the ICHYMOD hydrological model in predicting flood events in the Adige River basin.

**Author Contributions:** Conceptualization, S.S., M.Z. and M.B.; formal analysis, S.S.; investigation, S.S.; methodology, S.S., M.Z. and M.B.; project administration, F.G.; software, M.Z.; supervision, M.C., F.G. and M.B.; validation, S.S., M.Z., M.C., F.G. and M.B.; visualization, S.S.; writing—original draft, S.S. and M.B.; writing—review and editing, S.S., M.Z., M.C. and M.B. All authors have read and agreed to the published version of the manuscript.

**Funding:** The publication of this research was funded by NEXOGENESIS (grant agreement number 101003881), which is a 4-year European collaborative project financed by the European Commission under the H2020 programme, by the SECLI-FIRM project, and by the RETURN Extended Partnership and received funding from the European Union Next-GenerationEU (National Recovery and Resilience Plan–NRRP, Mission 4, Component 2, Investment 1.3–D.D. 1243 2/8/2022, PE0000005).

**Data Availability Statement:** ERA5 data used in this study are openly available in the Climate DataStore of the Copernicus Climate Change Service (https://cds.climate.copernicus.eu/#!/home, accessed on 5 May 2023).

**Acknowledgments:** This research was the part of the European Union's Horizon 2020 research and innovation program under grant agreement No 776868 (SECLI-FIRM project).

**Conflicts of Interest:** The authors declare no conflict of interest.

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
