# Peer review of "Scale Dependence of Errors in Snow Water Equivalent Simulations Using ERA5 Reanalysis over Alpine Basins"

_climate, doi:10.3390/cli11070154_

Round 1

Reviewer 1 Report

Review comments for “scale dependence of errors in snow water equivalent simulations using ERA5 reanalysis over alpine basins” by Shrestha et al.

This manuscript conducted the error analysis of the predicted snow water equivalent (SWE) using ERA-5 reanalysis temperature and precipitation as the input parameters, against the ground “truth” observations in the Alpine basins. It is found that the error to be basin-size dependent across the board. They also found correcting the ERA-5 precipitation bias alone only contributed slightly to improving the results, but correcting temperature bias helps improving the results decently. The bias correction is also found to be dependent on time for temperature but only a consistent offset for precipitation. The authors suggested in the end that correcting temperature bias as a function of time (separating in two periods in practice) works the best, but the scale-dependent error still exists. After the temperature and precipitation bias correction, the probability distribution of fractional area covered by snow matches well with the ground observations.

This work is well designed and executed. The amount of details put into the model descriptions is highly appreciated. As there are so many steps between the input temperature and precipitation and the output SWE, it’s extremely hard to diagnose and quantify the error sources. The sensitivity experiments designed in this manuscript concretely demonstrated the input error source, and provides a viable solution to utilize the reanalysis data. I encourage publication after handling some minor concerns.

Minor concerns (that I think need to be addressed before acceptation):

1. Section 2.4: I’m confused of the purpose of this ICHYMOD model. Is it planned to be used for the next step for flooding prediction, or it’s for providing the sink of your predicted SWE? Please clarify in this section and probably at the end of this paper when you mention your plan for the next step.

2. All the equations in this paper do not explicitly involve SWE parameter. Could you provide a high-level equation that list out the source and sink of SWE? Then you can break each term down into all equations in your paper.

3. Your sensitivity experiment is in-complete. Although it’s very unlikely to alter your conclusion, the changing-temperature-only experiment is needed (TC) in order to demonstrate that precip and temperature impact do not covary together but rather orthogonal.

4. It is visible that the temperature discrepancy does not come consistently in Fig. 2a, and that agrees with your conclusion. However, as I can see some “lagging” of the temperature timeseries between reference and ERA5, I’m wondering if it’s better to just use month-by-month correction factor to correct your ERA-5 temperature timeseries. This is indeed a normal procedure for dataset like IMERG, which is calibrated against ground rain gauges on a monthly basis to avoid drifting of the data record. In other words, why not enforce a more sophisticated calibration procedure?

5. I found Fig. 5 to be very interesting. Your prediction for Braies improves significantly after the temperature and precipitation correction, but for the Adige case (representing the mean situation in the basin), the discrepancy is visibly large at years when SWE peaks. Could you elaborate a bit more in the context what you think the cause would be? I’d trust to use the interannual variability for the Braies basin from your prediction, but not for Adige.

6. The current manuscript lacks a section dedicated to discussing other factors that could lead to the error in SWE prediction. It could be some physical processes that your models are missing, or some processes that are particularly sensitive to temperature (e.g., melting), or precipitation (e.g., mixed-phase precipitation seems to be not considered). It’s hard to quantify, but necessary to at least discussing them.

Reviewer 2 Report

The authors provided an important contribution about working with ERA5 data, in particular in Alpine environments. Knowing the relevance of the ERA 5 products, the findings can be useful for researcher in any snow dominated regions.

The article is well written. However, I have few comments, where the authors could consider more clarity:

1) In the abstract they write "...in 16 Alpine basins of varying catchment sizes...". Later we see that these basins are not independent but are subbasins of one larger basin, the upper Adige. Here the abstract is misleading. Actually, the model was evaluated in one catchment and the study investigated several subcatchments, which is fine as they look at scale issues. Maybe the abstract can be more clear in that point, as it implies that the study could have covered a larger spatial extend in the Alpine region. In reality, the flow of the larger subcatchments is influenced by what happens in the smaller headwaters.

2) The description of TOPMELT in the article repeats parts of the original paper by Zaramella et al. (2019), which is open access and thus could be easily referenced instead of duplicating the model description here.

3) The concept of bias correction is not fully convincing, or additional information is missing in the methodology description. Of course it is difficult to choose calibration and validation periods when variables have trends or breakpoints and it is a merit that the authors considered that point. But why are different strategies used for rainfall and for temperature? Isn't rainfall trended as well? Was it tested? The authors could more carefully differentiate here, e.g. first investigate if there are trends during the whole temporal model domain (in precipitation and temperature), then decide for the calibration and validation periods and a strategy for bias correction of climatic inputs. The coincidence of break point and calibration/validation period is good luck but it would be more convincing to include the test into a methodology/strategy.

4) The text from lines 305ff and the figure caption of figure 3 could be better matched. Without reading the text it is not understandable which are the bias corrected figures.

5) line 317: is there really a trend in KGE or better a correlation with watershed area?

All these comments do not necessarily require re-computations but I hope that the authors an improve the red line of their case study a bit.
